# Evaluation of Ecosystem Services in Ruoergai National Park, China

Hongfu Li, Yuelin Wang *, Wende Chen, Hongyu Li, Yue Tian and Ruojing Chen

College of Geography and Planning, Chengdu University of Technology, Chengdu 610059, China;
lhf000124cn@163.com (H.L.); chenwende@mail.cdut.edu.cn (W.C.); 19936640868@163.com (H.L.);
tianyuey0113@163.com (Y.T.); 13398352625@163.com (R.C.)
* Correspondence: wangyuelin001@163.com

**Abstract:** This study utilizes ArcGIS10.8 and ENVI5.3 software and the InVEST model to analyze and operate field survey data and remote sensing image data from Ruoergai National Park. The work simulates the soil retention, carbon storage, water supply, and habitat quality of the park to evaluate and analyze its four major ecosystem services. Subsequently, important areas of ecosystem services are zoned based on the results, aiming to provide decision-makers with a theoretical and scientific basis for formulating ecological environment restoration, protection, and management measures in Ruoergai National Park. The results showed the following: (1) In the study area, the land use types, ranked from largest to smallest, are grassland, unused land, forest, water area, and construction land. (2) Soil retention and water supply show an increasing trend, while carbon storage shows a decreasing trend. Habitat quality remains relatively stable, with most areas maintaining a high level of quality. (3) The importance zoning of ecosystem services in the study area exhibited a trend of "four increases and one decrease". Specifically, the areas classified as moderately important, highly important, and extremely important all increased, while the area designated as generally important decreased. The findings indicate that climate change, land use type changes, and human activities are the primary factors influencing changes in ESs. It is crucial to prioritize highly important and extremely important areas for protection and utilization within Ruoergai National Park. Moving forward, it will be essential to minimize human activities that disrupt the ecosystem, while also focusing on the conservation and sustainable use of forest and grassland.

**Keywords:** Ruoergai National Park; InVEST model; ArcGIS; ecosystem services; LULC

## 1. Introduction

Ecosystem services (ESs) are defined as the environmental conditions and processes provided by ecosystems to support and sustain human survival and development [1]. ESs are diverse and primarily revolve around supporting human life by providing essential resources like food and water [2]. These services also play a role in regulating climate and hydrology [3]. Additionally, ESs provide cultural services such as recreation and cultural amenities to people, while also contributing to soil erosion prevention [4]. Ecosystem service flows facilitate the transmission of ESs from ecosystems to human society. This is the dynamic process by which certain ESs move within different spatial regions over time, such as water flow, wind flow, and sand flow [5–7].

ESs have been extensively studied in recent decades [8,9]. Research on the dynamics of ecosystem services is no longer limited to static quantitative measures and necessitates a more profound exploration. For example, spatial indicators are used to evaluate ESs [10] and the trade-offs and synergistic relationships between ESs [11]. Furthermore, research has also examined the evaluation of ESs on various spatial scales [12,13].

In the late 20th century, the prominence of ESs rose and became a crucial consideration for socioeconomic development and human well-being. The evaluation of ESs has gradually gained recognition in forming ecological security patterns [14–16]. In 2014, the United Nations introduced an experimental framework for ecosystem accounting to integrate the

benefits and costs provided by ecosystems to humans into socioeconomic systems, highlighting the significance of ecosystems in promoting social development [17,18]. There is a diverse classification of ESs, as well as various evaluation methods and indicators [12,19]. However, there is currently no unified and standardized evaluation system in place. ES evaluation models include the InVEST model [20], ARIES model [21], SolVES model [22], and Enviro Atlas, EPM model. Among these models, the InVEST model is considered the most mature and advantageous [20]. InVEST is a GIS-based model that has been specifically developed to evaluate various ESs within landscapes across different land use scenarios [23]. Researchers have suggested that clarifying and evaluating ESs and integrating them into individual, corporate, and government decision-making processes can lead to the formulation of environmentally beneficial plans, ultimately achieving long-term harmonious development between humans and nature [24,25].

National parks have a vital role in China as protected natural areas, holding significant value in the ecosystem. These parks feature unique landscapes, cultural and natural heritage, and a variety of biological species [26]. The goal of establishing a national park system is to safeguard the authenticity and integrity of natural ecology and cultural heritage, protect biodiversity, and create a security barrier for the environment. It is a key element in constructing China's ecological civilization system [27]. Ruoergai National Park is situated in the transitional zone of the three natural regions of the country. It serves as a crucial water conservation and ecological functional area in the upper Yellow River basin. Known as the largest "plateau solid reservoir" globally, it also provides habitats for a diverse range of rare bird species. The park's diverse species play a crucial role in maintaining ecological balance, regulating climate, and conserving soil and water [28,29]. Therefore, assessing Ruoergai National Park's ecosystem services can help protect and utilize its rich natural resources, enhancing the region's overall ecological benefits.

This paper focuses on Ruoergai National Park as the research area and examines the land use change characteristics based on the area's basic data in different periods. The study utilizes the InVEST model to evaluate the ESs of the study area and identifies important areas for protection and utilization. These findings can assist decision-makers in developing plans for the protection of the area and provide a theoretical and scientific basis for park operation and management, ecological protection and restoration, and achieving the sustainable development of ecological and economic benefits. Additionally, the research contributes to the understanding of ecological service functions in alpine grassland meadow areas and provides a theoretical basis for further studies.

## 2. Study Area

The geographical coordinates of Ruoergai National Park (Sichuan area) range from 101°37′53.650″ E to 103°13′37.304″ E and 32°45′27.811″ N to 33°59′59.063″ N. Situated in the rift basin on the eastern edge of the Qinghai–Tibet Plateau, it is the confluence of the Qinling Mountains, Dieshan Mountains, and Minshan Mountains. The park is located in the northwest region of the Aba Tibetan and Qiang Autonomous Prefecture in Sichuan Province, at the junction between Sichuan, Gansu, and Qinghai Provinces. The study area is in the upper reaches of the Yellow River, administratively affiliated with Ruoergai County, Hongyuan County, and Aba County in Sichuan Province (Figure 1). The total area of the park is 13,829 km$^2$, with the Sichuan area covering 8336 km$^2$. It encompasses six natural reserves of various types, including four nature reserves and two national wetland parks. The park is particularly abundant in wetland resources, accounting for 91% of the swamp wetlands in the Ruoergai area. The landform of the study area consists of denuded hilly plateaus and high plains. The plateau landform is characterized by shallow hills and swamps, with intermittent hills and crisscrossing, winding ravines. These features result in extensive swamps and numerous oxbow lakes. The park is rich in river resources, being part of the Yellow River system, which includes the 174 km main stream of the Sichuan section of the Yellow River and three tributaries: Heihe, Baihe, and Jiaqu.

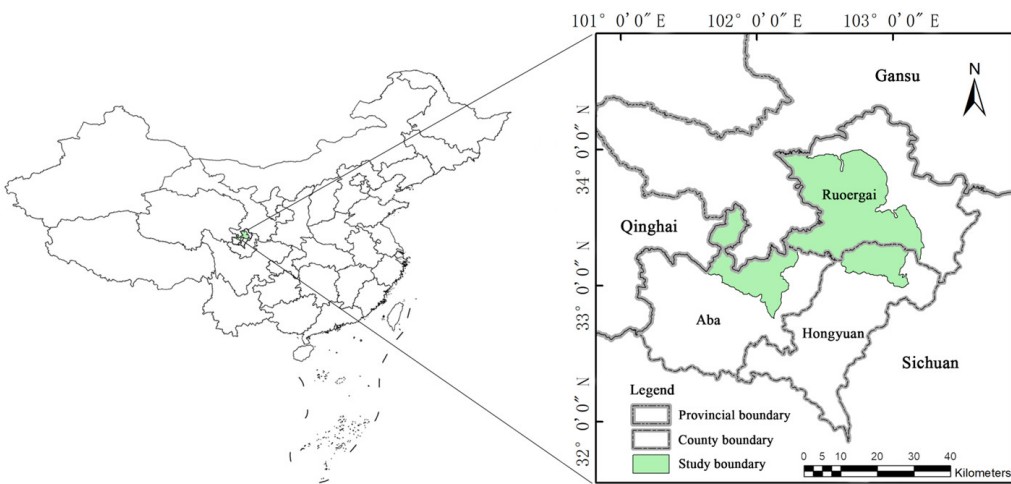

**Figure 1.** Schematic diagram of the location of the research area.

## 3. Material and Methods

### 3.1. LULC Image Acquisition and Processing

This article presents the land use classification images of Ruoergai National Park for the years 2010, 2015, and 2020. The original data consisted of remote sensing images obtained from the Landsat series data of China Geospatial Data (http://www.gscloud.cn/, accessed on 11 September 2023), with a resolution of 30 m and orbit numbers of 130/37 for all three phases. The process involved two major steps: data preprocessing and supervised classification. Preprocessing included band superposition, radiation calibration, atmospheric correction, and image cropping, all performed using ENVI software. The superimposed image was then radiation calibrated and subjected to atmospheric correction using the Flash module to eliminate errors caused by the remote sensing sensor and atmospheric scattering. Following preprocessing, ENVI's supervised classification was utilized for the interpretation and classification of the remote sensing images. The classification criteria of image land use type are shown in Table 1. The land in Ruoergai National Park was classified into five major types, construction land, forest, grassland, water area, and unused land, based on the national Grade I land classification standards and field conditions. Land use/cover images were generated from 2010 to 2020 (Figure 2).

**Table 1.** Image interpretation key for LULU in Ruoergai National Park.

| LULC | Image Interpretation Key |
|---|---|
| Forest | Including woodland, shrubbery, open woodland, and gardens. Distributed in patches along the ridge or mountainside. |
| Grassland | Irregular, patchy distribution; some areas are relatively fragmented, and the color is lighter than the forest. |
| Water body | Low-lying areas or low-altitude river valleys. Distributed in a strip or dot shape. |
| Construction land | Irregularly distributed in blocks, dots, and lines. The distribution is relatively concentrated, with clear boundaries. |
| Unutilized land | Mainly based on wetlands. Wetlands are characterized by low-lying terrain, inadequate drainage, and waterlogging. |

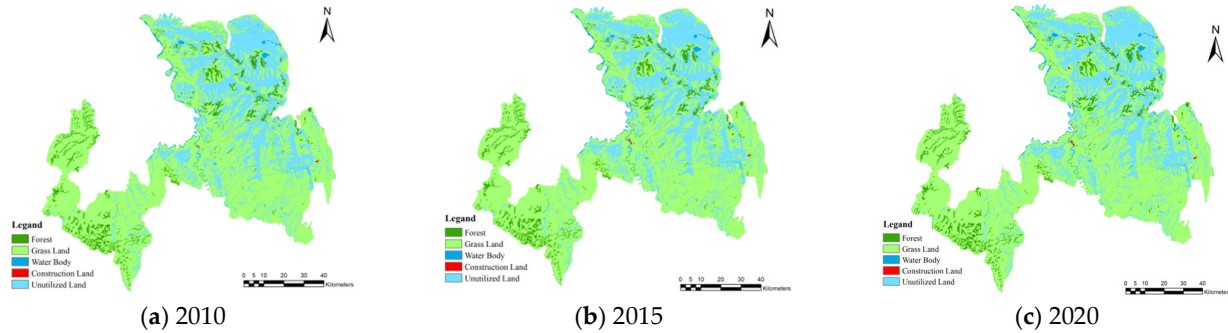

| (a) 2010 | (b) 2015 | (c) 2020 |

**Figure 2.** LULC of Ruoergai National Park from 2010 to 2020.

*3.2. Research Methods*

3.2.1. Soil Retention Model

(1)    Model principle

Currently, soil erosion quantitative models can be broadly categorized into two types: empirical statistical models and physical models. The USLE model is a well-known example of an empirical statistical model. In this study, we utilized the InVEST model in conjunction with the USLE model to assess soil conservation function in the study area for the years 2010, 2015, and 2020. The calculation formula used in this research is as follows [30]:

$$USLE = R \times K \times LS \times C \times P \tag{1}$$

$$RKLS = R \times K \times LS \tag{2}$$

$$SD = RKLS - USLE \tag{3}$$

where R represents the rainfall erosivity factor. K represents soil erodibility factor. LS represents the length and gradient of the slope. C represents vegetation coverage and management factors. P represents the factor of soil and water conservation measures.

(2)    Data source and processing

The main data required by the soil retention module of the InVEST model are shown in Table 2.

**Table 2.** Main data required for soil conservation in Ruoergai National Park.

| Data Type | Initial Data | Data Sources |
|---|---|---|
| DEM | DEM | Geospatial Data Cloud (http://www.gscloud.cn, accessed on 20 September 2023) |
| Annual and monthly average precipitation | Annual and monthly precipitation | National Earth System Science Data Center (http://www.geodata.cn, accessed on 20 September 2023) |
| Soil texture data | The maximum burial depth of soil roots, soil sand, silt, clay, organic carbon content, etc. | World soil database (Harmonized World Soil Database version 1.2, HWSD) |
| Vegetation coverage | NDVI data | Geospatial Data Cloud (http://www.gscloud.cn, accessed on 20 September 2023). Using the pixel binary model to calculate |
| LULC | Landsat image | Geospatial Data Cloud (http://www.gscloud.cn, accessed on 21 September 2023) |

3.2.2. Carbon Storage Model

(1)    Model principle

The carbon storage module of the InVEST model quantifies the amount of carbon dioxide stored in different carbon pools, including aboveground, underground, soil, and dead organic matter pools [31,32]. This article calculated the ecosystem carbon storage

based on land use conditions. First, the average carbon density of each land type's carbon pools was calculated. Then, this carbon density was multiplied by the area of each land type to obtain the total carbon storage value for the entire study area. The total carbon storage value was obtained by adding the carbon storage amounts of the four major carbon pools.

$$C_i = C_{i-above} + C_{i-below} + C_{i-soil} + C_{i-dead} \tag{4}$$

$$C_{total} = \sum_{i=1}^{n} C_i \times S_i \tag{5}$$

where $i$ is a certain land use type, and $C_i$ is the total carbon density of the LULC. $C_{i-above}$, $C_{i-below}$, $C_{i-soil}$, $C_{i-dead}$, and $C_{total}$ are the corresponding LULC's vegetation carbon density, live underground root of carbon density, the density of carbon in the soil, vegetation litter of carbon density, and total carbon storage. $S_i$ represents the total area of the LULC. $n$ shows the sum of the ways of using the land.

(2) Data source and processing

The carbon density data were derived from the "Chinese Soil Data Set Based on the World Soil Database" and referred to the research results of some scholars [33,34]. The results are shown in Table 3.

**Table 3.** Carbon pool density values of different LULCs in the study area (t/h).

| LULC | C_Above | C_Below | C_Soil | C_Dead |
|---|---|---|---|---|
| Forest | 66.55 | 11.31 | 242.13 | 0 |
| Grassland | 0.42 | 11.16 | 215 | 0 |
| Water body | 0 | 0 | 0 | 0 |
| Construction land | 0.59 | 0 | 78.49 | 0 |
| Unutilized land | 0 | 0 | 71.45 | 0 |

### 3.2.3. Water Supply Model

(1) Model principle

The water source supply in the InVEST model was determined by calculating the difference between the rainfall and the actual evaporation for each grid unit in the watershed. These values were then added together to calculate the total water source supply for the watershed. The basic formula for this calculation is as follows [35–37]:

$$Y_x = P_x - AET_x \tag{6}$$

where $Y_x$ is the annual water supply of grid cell $x$ (mm). $AET_x$ represents the actual evapotranspiration of cell $x$. $P_x$ is the average annual precipitation (mm) of cell $x$.

(2) Data source and processing

Precipitation, potential evapotranspiration, LULC, elevation, soil properties, and other data sources and corresponding treatments are shown in Table 4. In addition, the $Z$ parameter was inversely proportional to the water production volume and required correction based on the statistical analysis of the results. The calibration method involved using the average water production and annual precipitation to calculate the water production coefficient. If the calculated coefficient aligned with the one in the water resources bulletin of the study area, the $Z$ parameter was considered appropriate. However, if the calculated coefficient was lower than the one reported in the bulletin, water production needed to be increased, leading to a reduction in the $Z$ parameter. This process needed to be repeated until the model yielded satisfactory results.

**Table 4.** Main data required for water supply in Ruoergai National Park.

| Data | Source and Treatment |
|---|---|
| Annual average precipitation, potential evapotranspiration | National Earth System Science Data Center (http://www.geodata.cn, accessed on 26 September 2023) |
| LULC | Geospatial Data Cloud (http://www.gscloud.cn, accessed on 21 September 2023) |
| DEM | Geospatial Data Cloud (http://www.gscloud.cn, accessed on 20 September 2023). Calculate the percentage Slope using the Slope tool |
| Soil properties (soil thickness, soil texture, organic matter content, etc.) | World soil database (Harmonized World Soil Database-HWSD version 1.2). Soil texture was used to calculate plant water availability and soil saturated water conductivity |
| Others (maximum root depth, vegetation evapotranspiration coefficient, velocity coefficient) | References research results, FAO crop reference values and InVEST model documentation |

### 3.2.4. Habitat Quality Model

(1)　Model principle

The InVEST habitat quality module performs comprehensive calculations by combining the sensitivity of landscape types and the intensity of external threats. It takes into account factors such as the influence distance and spatial weight of stress factors. The module treats habitat quality as a continuous variable and fully considers it in assessments. It also studies the impact of changes in land cover patterns on habitat quality [38–40]. The calculation is as follows:

$$Q_{xj} = H_j \left[ 1 - \frac{D_{xj}^z}{D_{xj}^z + k^z} \right] \tag{7}$$

where $Q_{xj}$ is the habitat quality index of grid $x$ in land use type $j$. $H_j$ is the habitat suitability of habitat type $j$. $k$ is a half-full sum constant, usually set to 0.5. $Z$ is a normalized constant.

(2)　Data source and processing

①　Threat factor data

This paper integrated the InVEST model user manual with existing data, taking into account previous research findings and the specific characteristics of the study area. It identified four land types, namely, urban land, rural settlements, construction land, and bare land, as threat factors (Table 5). The maximum influence distance and weight were assigned to determine the recession type of each land type.

②　Habitat adaptation and sensitivity data

The habitat quality of each land use type was determined by its own habitat suitability and sensitivity to various threat factors. In general, natural and complex systems have higher habitat suitability, while artificial systems have no habitat suitability [41,42]. The degradation of an LULC is greater when it is more sensitive to threatening sources.

**Table 5.** Maximum threat distance, weight, and attenuation type of threat sources.

| Threat | Max_Dist/km | Weight | Decay |
|---|---|---|---|
| Urban land | 8.0 | 1 | Exponential |
| Rural settlements | 5.0 | 0.6 | Exponential |
| Construction land | 6.0 | 0.7 | linear |
| Bare land | 2.5 | 0.3 | Exponential |

### 3.2.5. Ecosystem Services Importance Zone

This paper utilized the quantile classification method to propose a novel classification of the ESs of Ruoergai National Park, including its overall ESs. The study employed membership function analysis and the range standardization method to ensure consistency in the dimensions of various ecological service functions. The raster layers of each ecological service function were overlaid and summed, followed by an evaluation of the importance of each function. Based on their degree of importance, the functions were categorized into four levels: generally important, moderately important, highly important, and extremely important. This was carried out to better study the current status of ESs in a quantitative and spatialized expression.

## 4. Results and Analysis

### *4.1. LULC Change Analysis*

#### 4.1.1. Temporal and Spatial Characteristics of LULC

According to the LULC diagram of the third phase of Ruoergai National Park, the area of the five major land types was calculated using the regional analysis function in ArcGIS software. The results are presented in Table 6 below. Over time, there have been varying degrees of changes in the area of all land types within Ruoergai National Park, as illustrated in Figure 3. In general, the LULC in Ruoergai National Park, ranked from largest to smallest, are grassland, unused land, forestland, water, and construction land. The study area is characterized by high mountains and deep valleys. Woodlands and grasslands are predominantly found in the southern, western, and southeastern regions, which are mountainous. Unused land is concentrated in the northern, central, and southeastern areas, with some also present in the southwest. Water bodies are mainly distributed in the northern and central regions, with scattered water areas near construction land in the east. Construction land is primarily located in the central and eastern areas, aligning with the distribution of water areas. Analyzing the changes in the area of different landscape types, it can be observed that the area of woodland and water bodies has remained relatively stable. However, water areas have experienced significant shrinkage, with a decrease of 1.67 km$^2$. Grasslands have shown an initial increase followed by a decrease, with a decrease of 6.66 km$^2$ from 2015 to 2020. On the other hand, the area of construction land and unused land continues to expand.

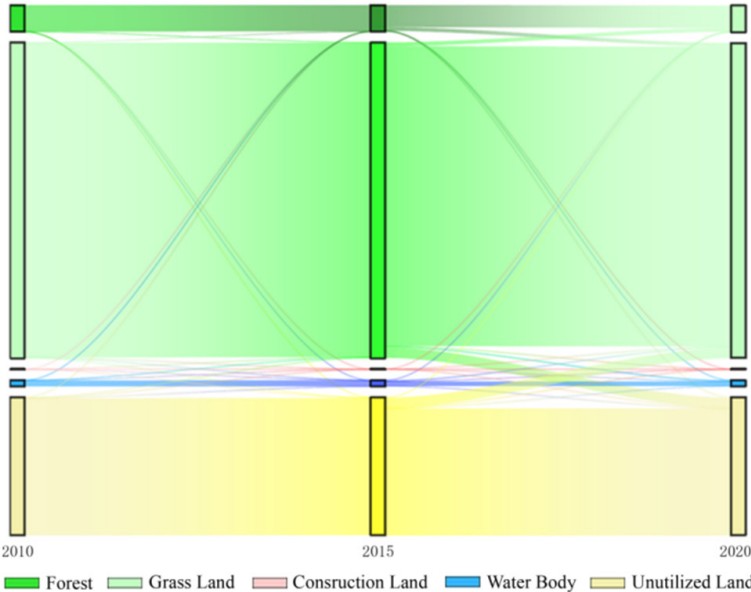

**Figure 3.** Sankey diagram of landscape transfer in Ruoergai National Park from 2010 to 2020.

**Table 6.** Proportion of LULC and areas in Ruoergai National Park from 2010 to 2020.

| LULC | 2010 | | 2015 | | 2020 | |
|---|---|---|---|---|---|---|
| | Area/km$^2$ | Proportion | Area/km$^2$ | Proportion | Area/km$^2$ | Proportion |
| Forest | 456.81 | 5.48% | 455.98 | 5.47% | 456.81 | 5.48% |
| Grassland | 5400.06 | 64.78% | 5400.89 | 64.79% | 5394.23 | 64.71% |
| Water body | 110.87 | 1.33% | 108.37 | 1.30% | 109.20 | 1.31% |
| Construction land | 8.08 | 0.09% | 8.60 | 0.10% | 11.62 | 0.14% |
| Unutilized land | 2360.76 | 28.32% | 2362.42 | 28.34% | 2364.09 | 28.36% |

### 4.1.2. Analysis of LULC Transfer Change

Based on the Sankey diagram (Figure 3), it is evident that there were shifts between different landscape types from 2010 to 2015. However, the overall distribution of land use did not undergo significant changes. From 2015 to 2020, there was a minor exchange between woodland and grassland, while the exchange between grassland and unused land was more pronounced. In general, there were frequent transfers between different landscape types, but their respective areas remained relatively stable.

### *4.2. Ecosystem Services Assessment*

### 4.2.1. Soil Retention Assessment

Statistical analysis was conducted on the potential soil erosion amount, actual soil erosion amount, and soil conservation in three periods (2010, 2015, and 2020) in Ruoergai National Park. The overall potential soil erosion amount in these years was calculated to be $7.47 \times 10^5$ t, $5.84 \times 10^5$ t, and $9.54 \times 10^5$ t, respectively. The actual soil erosion amounts were $8.76 \times 10^4$ t, $6.79 \times 10^4$ t, and $1.14 \times 10^5$ t, respectively, while the soil retention amounts were $6.59 \times 10^5$ t, $5.16 \times 10^5$ t, and $8.40 \times 10^5$ t, respectively. Overall (Figure 4), the soil conservation capacity of grassland and unused land with a large area is at a low level, whereas the soil conservation capacity of woodland is at a high level. The total amount of soil conservation showed a trend of first decreasing and then increasing from 2010 to 2020. The soil conservation amount was the highest in 2020, with an increase of $1.81 \times 10^5$ t from 2010. Among them, forestland has the highest soil conservation capacity. The decrease in soil conservation capacity in 2015 is attributed to the reduction in forestland area and the expansion of construction land. However, later, in response to various national environmental protection policies, the Ruoergai Wetland was protected and restored, leading to an increase in soil conservation capacity.

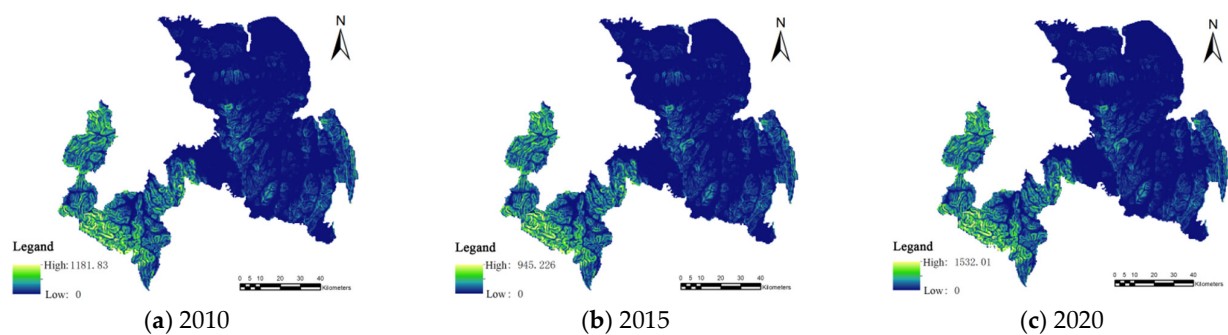

(**a**) 2010        (**b**) 2015        (**c**) 2020

**Figure 4.** Spatial distribution of soil conservation in Ruoergai National Park from 2010 to 2020.

From a spatial distribution perspective, the overall soil conservation in the study area exhibits a distribution pattern of being "high in the west and low in the east". This spatial distribution characteristic of soil conservation is influenced by altitude, as well as factors such as climate and precipitation in the study area. The areas with median and high soil conservation values are concentrated in regions with lower altitudes, primarily characterized by woodland land use. Woodland and shrubland play a significant role in

retaining soil and reducing soil loss. On the other hand, areas with low soil conservation values are more prevalent in higher-altitude areas.

### 4.2.2. Carbon Storage Assessment

Statistical analysis was conducted to determine the carbon reserves in Ruoergai National Park during three periods. The average carbon densities in 2010, 2015, and 2020 were calculated to be approximately 16.61 t/hm$^2$, 16.62 t/hm$^2$, and 16.61 t/hm$^2$, respectively. The overall carbon reserves were approximately 1580.66 $\times$ 10$^5$ t, 1580.78 $\times$ 10$^5$ t, and 1579.13 $\times$ 10$^5$ t. Over the span of 2010 to 2015, the total carbon storage increased by 0.12 $\times$ 10$^5$ t, while from 2015 to 2020, it decreased by 1.65 $\times$ 10$^5$ t. In a span of 10 years, the total carbon storage decreased by 1.53 $\times$ 10$^5$ t. Figure 5 illustrates the trend of carbon reserves from 2010 to 2020, showing an initial increase followed by a decrease. Forestland exhibited the highest carbon storage among the different ecosystems. The study area, located on the Qinghai–Tibet Plateau, experiences temperature-driven effects on the soil organic carbon cycle in meadow grassland and swamp ecosystems. Global warming is leading to permafrost melting, accelerating plant root production and turnover rates [43], the decomposition of fallen leaves [44,45], and changes in microbial community structure [46], resulting in increased $CO_2$ release. As atmospheric $CO_2$ concentration rises, plants' photosynthesis and productivity increase, leading to greater input of carbon into soil carbon pools [47]. Some studies have found that under warm and humid climates, alpine plants absorb carbon at a faster rate than frozen soil releases it. The carbon storage of ESs has weakened due to the expansion of construction land caused by intensified human activities. Although the area of unused land has increased, there has been a greater decrease in grassland area, which has resulted in a decrease in carbon storage in Ruoergai National Park.

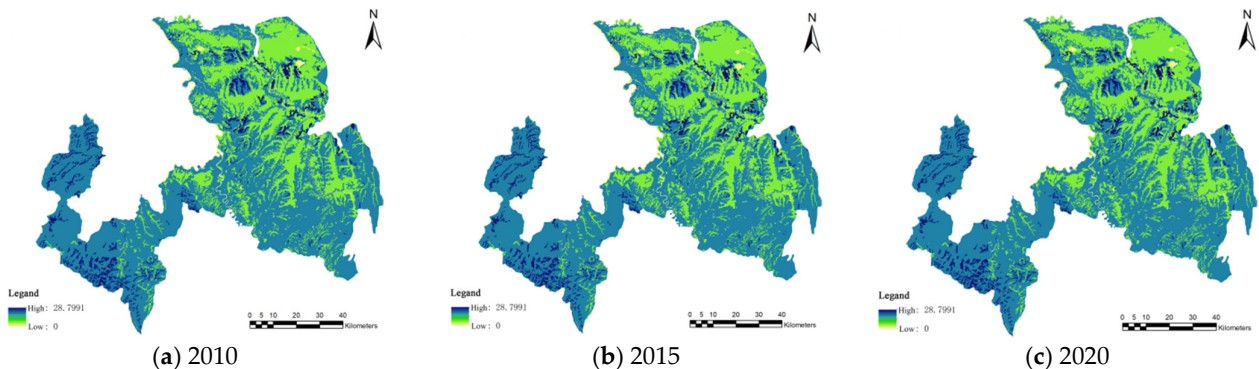

(**a**) 2010         (**b**) 2015         (**c**) 2020

**Figure 5.** Spatial distribution of carbon storage in Ruoergai National Park from 2010 to 2020.

The spatial distribution of carbon storage in the study area has remained relatively unchanged over the past 10 years. The highest unit carbon storage value is 28.7991 t C/hm$^2$. The overall spatial distribution pattern shows that carbon storage is high in the west and low in the east, and low in the north and high in the south. The distribution of carbon storage is closely related to land use type, and climate and precipitation in the study area also have an impact. Figure 5 shows that the areas with high carbon storage values are mainly concentrated in low-altitude areas in the south, with forestland as the main land use type. The areas with median carbon storage values are mainly concentrated in high-altitude areas in the north and central areas, with grassland and unused land as the main land use types. The areas with low-value carbon storage are mostly composed of water bodies and construction land.

### 4.2.3. Water Supply Function Assessment

A statistical analysis was conducted to determine the water production in the three phases. Specifically, the water production in Ruoergai National Park was calculated for the years 2010, 2015, and 2020, yielding values of approximately 2.39 $\times$ 10$^6$ mm, 1.78 $\times$ 10$^6$ mm, and 3.18 $\times$ 10$^6$ mm, respectively. Additionally, the average water production in the

study area for the same years was found to be 287.10 mm, 213.31 mm, and 381.99 mm, respectively. Over the period from 2010 to 2020, the water production exhibited a pattern of initially decreasing, followed by an increase, ultimately resulting in a net increase of $7.9 \times 10^5$ mm. Moreover, it is evident that the precipitation intensity has an impact on the water supply capacity.

From a spatial scale perspective, the water supply in the study area exhibits a distribution pattern characterized as "high in the east, middle in the west, high in the north, and middle in the south" (Figure 6). This spatial distribution is closely linked to actual evaporation, climate, and precipitation. Areas with lower evapotranspiration tend to have greater water supply in regions with higher precipitation levels. The high-value water source supply areas are primarily located in the northern and central regions, characterized by higher altitudes. The land use in these areas is predominantly unused land, with wetlands being the most common type. Additionally, the melting of ice, snow, and glaciers on the Qinghai–Tibet Plateau also contributes significantly to the water sources. The medium-value water supply areas are mainly found in the western and southern parts of the study area, with woodland and grassland being the dominant land use types, resulting in diverse vegetation. On the other hand, low-value water supply areas are scattered throughout the study area and consist mainly of water bodies, sandy land, and bare land.

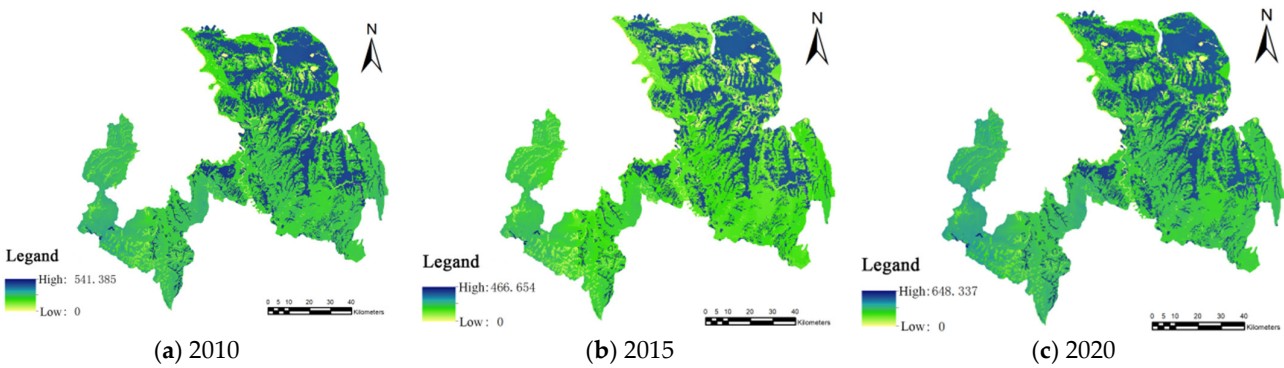

(**a**) 2010      (**b**) 2015      (**c**) 2020

**Figure 6.** Spatial distribution of water supply in Ruoergai National Park from 2010 to 2020.

4.2.4. Habitat Quality Assessment

The habitat quality index is a comprehensive measure used to evaluate the quality of habitat based on the land use conditions in the study area [48]. The average habitat quality index in the study area was 0.7036 in 2010, 0.7033 in 2015, and 0.7042 in 2020. The overall habitat quality has been gradually improving and remains generally stable. Figure 7 illustrates that the habitat quality is relatively high in most areas of the study area, particularly in the south and west where there is abundant vegetation and high forest and grass coverage. The central and northern parts of the area also have relatively good habitat quality, mainly due to the extensive coverage of hygrophytes. The improvement in habitat quality from 2010 to 2020 can be attributed to the implementation of various conservation policies and measures, such as large-scale wetland ecological restoration, land desertification control, and water ecological protection and restoration in the study area in recent years.

From a spatial scale perspective, the overall habitat quality of the study area exhibits a distribution pattern characterized by being "high in the west and low in the east, low in the north and high in the south". The areas with high habitat quality values are primarily concentrated in the mountainous regions of the west and south, and these areas also coincide with the high-value areas of the Digital Elevation Model, indicating that they have experienced less erosion due to human activities. The dominant vegetation coverage types are predominantly woodland and grassland, and the different ecosystems intertwine, resulting in a rich and diverse range of species. The areas with median habitat quality are concentrated in the northern and central regions, with the main land type being swamp wetland in unused land, which harbors a relatively abundant species composition. On the

other hand, the areas with median habitat quality and construction land are subject to frequent human activities, and they exhibit a high level of similarity with bare land and sandy land, indicating significant impacts from human activities. As a result, the naturalness and integrity of the ecological environment in these areas are relatively compromised.

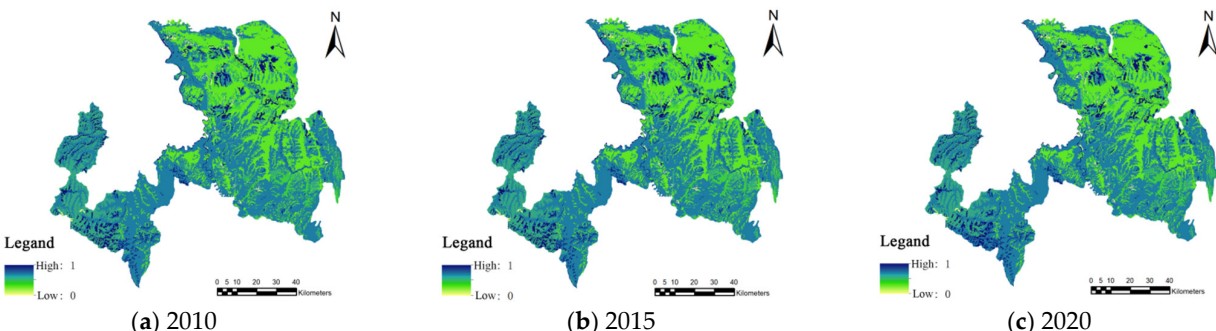

(**a**) 2010      (**b**) 2015      (**c**) 2020

**Figure 7.** Spatial distribution of habitat quality in Ruoergai National Park from 2010 to 2020.

4.2.5. Ruoergai National Park Ecosystem Service Importance Zones

As shown in Table 7, the overall fluctuation in the importance of ESs in the study area from 2010 to 2020 is not significant. The trend of change is "four increases and one decrease". The areas of moderately, highly, and extremely important grades increased by 3.47%, 7.67%, and 9.16%, respectively. The area of the generally important grade decreased significantly, with a decrease of 20.3%. By 2020, the generally important area of ESs of Ruoergai National Park had a very small distribution area of 419.37 km², accounting for only 5.03% of the park area. This district is mainly distributed in the low-elevation depression area, and ESs are poor.

**Table 7.** Area proportion of ecological service function importance zones in Ruoergai National Park from 2010 to 2020.

| Importance Zoning | 2010 | | 2015 | | 2020 | |
|---|---|---|---|---|---|---|
| | Area (km²) | Proportion | Area (km²) | Proportion | Area (km²) | Proportion |
| Generally important | 2111.29 | 25.33% | 2674.36 | 32.08% | 419.37 | 5.03% |
| Moderately important | 3033.90 | 36.40% | 2501.80 | 30.01% | 3323.89 | 39.87% |
| Highly important | 1669.61 | 20.03% | 1580.02 | 18.95% | 2308.95 | 27.70% |
| Extremely important | 1521.60 | 18.24% | 1579.83 | 18.94% | 2283.78 | 27.40% |

Table 7 shows the order of the proportion of ESs in the study area: moderately important (35.43%) > highly important (22.23%) > extremely important (21.53%) > generally important (20.81%). From Figure 8, it can be observed that the generally and moderately important areas are widely distributed in the north, central, and east of the study area. The predominant land use types in these areas are unused land, grassland, water, and construction land. However, these areas have poor soil conservation capacity and relatively low carbon sequestration capacity. They also exhibit high water supply but low habitat quality. Therefore, they are not considered of high importance in the study area. It is necessary to treat the existing land to prevent further desertification and enhance vegetation richness. Native tree species with strong pest resistance and resistance to barrenness are preferred. On the other hand, the highly important areas are mainly located in the east and south of the study area, characterized by high-coverage grasslands. These areas have a favorable ecological environment and strong ecosystem service capabilities. Lastly, the extremely important areas are primarily found in the low-altitude areas of the Qilian Mountains, with a small portion distributed in high-altitude areas. The land use in these areas is mainly woodland, and they experience minimal ecological disturbance. Most of these areas consist of native natural landscapes.

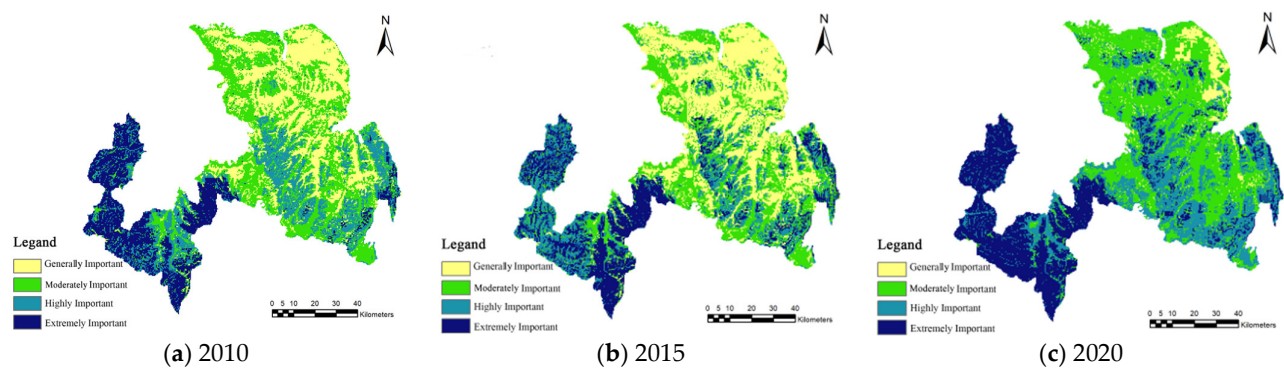

**Figure 8.** Importance zoning of ESs in Ruoergai National Park from 2010 to 2020.

The two levels of highly important and extremely important are crucial aspects of ESs in the study area. These aspects play a vital role in regulating regional climate and hydrology, sequestering carbon, providing subsistence supply, nutrient recycling, and maintaining biodiversity. If the ecosystem in this area is severely damaged, it will have a significant negative impact on the ecological barrier function and sustainable development of the environment in the entire study area. Therefore, it is essential to restrict human activities reasonably, protect natural resources, and utilize them scientifically while maintaining a complete ecosystem network.

## 5. Discussion

### 5.1. Drivers Affecting Ecosystem Services in Ruoergai National Park

Ecological systems are crucial for human survival and have a significant impact on regional sustainable development. Ruoergai National Park, as a key ecological functional area, plays a vital role in protecting ecosystem health, promoting ecological restoration, and maintaining functional stability. This study examines the changing trends, influencing factors, and important functional divisions of ESs in the area. The findings indicate that from 2010 to 2020, Ruoergai National Park witnessed an upward trend in soil conservation, water conservation capacity, and habitat quality, but a downward trend in carbon storage capacity. Recent years have seen increased rainfall, low temperatures, and a cold and humid climate in the Ruoergai area, which has positively contributed to improved water conservation capabilities. The study area has a significant amount of unused land, including wetlands and glaciers in the northern and central parts, as well as extensive grassland and woodland. The vegetation conditions overall are favorable, contributing to improved soil retention capacity and habitat quality. However, human activities have led to an increase in construction land and a decrease in grassland, thereby hindering the improvement in carbon storage capacity. Since the establishment of the Ruoergai Wetland National Nature Reserve in 1998, measures such as wetland ecological restoration, land desertification control, and water ecological protection and restoration have effectively mitigated ecological degradation. Changes in climate conditions and human activities are the driving forces behind the spatial distribution changes in ESs in Ruoergai National Park.

### 5.2. Strategies to Restoration Grasslands in the Future

Continued efforts should be made to strengthen the implementation of grassland ecological restoration projects. Firstly, prioritizing the preservation of native wild grass species resources and establishing a technical system for the domestication, selection, breeding, and expansion of these species is crucial. Secondly, it is important to adopt appropriate engineering restoration measures that are tailored to local conditions. Lastly, establishing pilot projects for the restoration of degraded grasslands and developing specific and targeted protection plans is necessary, such as grassland fencing [49]. Simultaneously, raising awareness about ecological protection is essential. Regional and departmental management agencies should focus on improving the institutional system of grassland education, orga-

nizing various publicity and public activities to promote people's understanding of and commitment to grassland protection. In the future, the restoration of degraded grasslands should integrate new technologies and materials from the field of ecological restoration, fostering innovation and forming a diverse and integrated technical system. This approach will enable the refined management of ecological restoration in degraded grasslands.

### 5.3. Limitations and Future Perspectives

In order to analyze the reasons for the changes in the spatial and temporal distribution of ESs, this article conducted a detailed analysis of each element of the evaluation process. The spatial distribution of ES is primarily influenced by climate conditions, land use changes, and human activities. Additionally, ecological sensitivity is also an important factor that affects ecosystem evaluation. It primarily represents the sensitivity of the ecosystem to natural and human activities in the evaluation area, which can be observed through indicators such as soil erosion and rocky desertification [50]. By combining ecological sensitivity and ecological function importance in the spatial evaluation results, a more scientifically formulated ecological protection and restoration system for Ruoergai National Park can be developed.

However, this study has some limitations as it only focused on four typical ESs. The parameter setting of the model was based on relevant research and continuous debugging, which may have introduced subjective factors. To enhance the protection and utilization of Ruoergai National Park, it would be beneficial to include more comprehensive ESs, such as biodiversity assessment and water quality purification, in future research. Additionally, the time range of this study was limited to 2010–2020, which may not have captured longer-term changes in land use and ESs. In future research, we can delve into the trade-offs and synergies between ESs. By proposing more targeted and accurate ecological protection policies, we can provide a scientific basis for coordinating ESs and promoting a balanced development of ecology and the social economy.

### 6. Conclusions

This article utilizes the InVEST model to analyze the changing trends, spatiotemporal distribution, and influencing factors of soil conservation, carbon storage, water supply, and habitat index in Ruoergai National Park from 2010 to 2020. The results reveal the following:

From 2010 to 2020, the overall land use types in Ruoergai National Park ranked in the following order from largest to smallest: grassland > unused land > forest > water body > construction land. The transitions between land types have remained relatively stable with no significant changes. The areas of grassland and water have decreased, while forest land and unused land have been the main recipients of these transitions. Conversely, the areas of construction land and unused land have increased, with grassland and water being the primary sources. The grassland area experienced an upward trend from 2010 to 2015, followed by a downward trend from 2015 to 2020, resulting in a total decrease of 5.83 km$^2$ in the grassland area.

Climatic conditions, land use changes, and human activities are the primary factors influencing changes in ESs. Between 2010 and 2020, the total soil conservation capacity of Ruoergai National Park initially decreased, then increased, and finally showed an overall increase of $1.81 \times 10^5$ t. The soil conservation capacity improved, particularly in forested areas. The distribution of soil conservation exhibits a spatial pattern of being "high in the west and low in the east". During the same period, the total carbon storage in Ruoergai National Park initially increased, then decreased, and ultimately exhibited a decreasing trend, with a total decrease of $1.53 \times 10^5$ t. The carbon storage declined, despite the presence of diverse forestland vegetation types with the highest carbon storage. The overall carbon storage in the study area follows a spatial distribution pattern of being "high in the west and low in the east, low in the north and high in the south". From 2010 to 2020, the total water conservation amount in Ruoergai National Park initially decreased, then increased, and finally showed an increasing trend, with an increase of $7.9 \times 10^5$ mm. The

water conservation amount improved. Unused land exhibited high water production, while forest land and grassland had moderate levels, and water areas and construction sites showed low water production. The overall water supply in the study area demonstrates a spatial distribution pattern of being "high in the east, moderate in the west, high in the north, and moderate in the south". Between 2010 and 2020, the habitat quality of Ruoergai National Park remained relatively stable. Most areas have relatively high habitat quality, although the northern, central, and some eastern parts experienced relatively high levels of habitat degradation. The overall habitat quality in the study area exhibits a spatial distribution pattern of being "high in the west and low in the east, low in the north, and high in the south".

From 2010 to 2020, the importance of ESs in the study area exhibited a pattern of "four increases and one decrease". This means that the area of moderately important, highly important, and extremely important areas increased, while the area of generally important areas decreased. The order of the proportion of ESs in the study area is as follows: moderately important > highly important > extremely important > generally important. By 2020, the total distribution area of highly important areas and extremely important areas for ESs in Ruoergai National Park amounted to 4592.73 km$^2$, which accounted for half of the total area of the study area. The predominant land use types in this area are woodland and grassland, which have optimal ecological system service functions. The total area of moderately important areas with ESs in the study area is 3323.89 km$^2$, accounting for approximately 40% of the total area. The land use type in this area is mainly unused land. Therefore, in the next planning and policy formulation process, greater attention should be given to the protection and utilization of land types such as forest land, grassland, and unused land.

**Author Contributions:** Conceptualization, H.L. (Hongfu Li), Y.W. and W.C.; methodology, H.L. (Hongfu Li), Y.W. and W.C.; software, H.L. (Hongfu Li); validation, H.L. (Hongyu Li); formal analysis, H.L. (Hongfu Li) and Y.T.; data curation, H.L. (Hongfu Li) and R.C.; writing—review and editing, H.L. (Hongfu Li); supervision, Y.W. and W.C. All authors have read and agreed to the published version of the manuscript.

**Funding:** This work was supported by the Sichuan Provincial Philosophy and Social Science Research "14th Five-Year Plan" 2022 Annual Project: Evaluation of Ecosystem Services in Ruoergai National Park (No. SC22BZD075).

**Institutional Review Board Statement:** Not applicable.

**Informed Consent Statement:** Not applicable.

**Data Availability Statement:** The datasets generated and analyzed during this study are not publicly available, but can be obtained from the authors upon reasonable request.

**Conflicts of Interest:** The authors declare no conflicts of interest.

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
