# Peer review of "Evaluation of Ecosystem Services in Ruoergai National Park, China"

_sustainability, doi:10.3390/su16083241_

Round 1

Reviewer 1 Report

Comments and Suggestions for Authors

Check the capital in the sentence well.

Acronym needs to be used after presenting full name first. 

What could become the stream of the eco system service?

What could become the definition in the extant literature.?

It is not presented in the research well. 

The visibility of figure 1 very poor. Authors need to present the explanation for each figure and table more concretely.

The paper needs to present theoretical contribution more strongly.

Also, authors need to present more about the data and how the analytic tool is adequate to reach the conclusion. 

Authors need to present the limitation and directions for future research at the end of the manuscript. 

Reviewer 2 Report

Comments and Suggestions for Authors

In the review of your manuscript, the following sections have been identified for revision:

This would shorten the abstract in summarizing the objective, methodology, key findings, and implications of the study further. The significance of this research study and how the results obtained from the study help in contributing towards the value of the research should be articulated properly.

in introduction section, please consider provide a more in-depth background for the relevance of the ecosystem framework services, and in this part, clearly outline its relevance to Ruoergai National Park, which will be used for your study.

the manuscript should include detailed explanations in relation to the selection and calibration of InVEST model parameters. Transparency and reproducibility require justification of model choice and any modifications made for this study area and therefore only this sub-section needs to be revised.

Besides, it should be clear that the methodological approach for the classification of land use and the criteria to distinguish different types of land. 

Comments on the Quality of English Language

it appears that the text demonstrates a good command of English, with only occasional issues that might need minor revisions for clarity, grammar, or punctuation

Reviewer 3 Report

Comments and Suggestions for Authors

The article's subject is timely and interesting for research. It aligns with the aim and scope of the Sustainability journal's special issue Ecosystem Services and Human Wellbeing: Linking Science, Policy, and Practices.

The article generally has a logical and acceptable structure, but some parts need improvement.

The abstract is too long (it must be a maximum of 200 words) and poorly structured. It should be reviewed and rewritten more concisely. I suggest revising the abstract to adopt the following standard abstract structure: Background, Purpose, Methodology, Results/Conclusions, Contributions/Implications.

Although the Introduction section briefly introduces the background and specifies the article’s aim, it would be preferable to reference the specialized literature, present the main methods and results, and identify the scientific gap. The link to the literature analysing such a topic would also be helpful in the discussion section.

Overall, I found this work interesting and beneficial.

Comments on the Quality of English Language

Acceptable

Round 2

Reviewer 1 Report

Comments and Suggestions for Authors

It is revised well.

Reviewer 2 Report

Comments and Suggestions for Authors

Dear authors,

After evaluating the revised manuscript, I identify significant improvements. Specifically, the abstract has been enhanced to concisely highlight the study objectives, methodology, key results, and relevance. Furthermore, the introduction now provides a deeper background on the relevance of ecosystem services in the context of Ruoergai National Park. The methodology presented in the most recent version has also improved, with a clearer explanation of the selection and specific restrictions of the InVEST model, which contributes to the transparency and reproducibility of the study.

Based on the above, I recommend the current version for publication.

Yours sincerely,